# Gastrointestinal Microorganisms and Blood Metabolites in Holstein Calves with Different Heat Stress Responses in the Same Hot Environment

**DOI:** 10.3390/microorganisms13040801

**Published:** 2025-03-31

**Authors:** Zhanhe Zhang, Guangrui Zhao, Song Niu, Yang Jia, Donglin Wu, Ming Xu

**Affiliations:** 1College of Animal Science, Inner Mongolia Agricultural University, Hohhot 010018, China; 506@emails.imau.edu.cn (Z.Z.); dk506@emails.imau.edu.cn (G.Z.); ndns@emails.imau.edu.cn (S.N.); jiayang@imau.edu.cn (Y.J.); 2Baotou Beichen Feed Technology Co., Ltd., Baotou 014000, China

**Keywords:** calves, heat stress response, gastrointestinal microorganisms, serum metabolome

## Abstract

This study investigated differences in gastrointestinal microbiota and blood metabolomes in calves with different heat stress (HS) responses in the same hot environment. Ten high (H) and ten low (L) HS response preweaning Holstein calves were selected based on their heat stress level (respiratory rate and rectal temperature), jugular vein blood samples and ruminal and fecal samples were collected. Notable variations were observed in the serum levels of heat shock protein 70 (HSP-70) and IL-2 between the two calf groups (*p* < 0.05). In group H, rumen and fecal microbiota synergism was disrupted. In the H group, the host metabolome exhibited enrichment in pyruvate metabolism and the tricarboxylic acid cycle (*p* < 0.05). Key factors bridging the relationship between gastrointestinal microbiota and serum metabolites included the rumen bacterial genus *g__Ruminococcus*, serum HSP-70, malic acid, and fumaric acid. These hubs served as potential indicators for distinguishing the response to heat stress in calves (*p* < 0.05). In conclusion, this study identified the relationship between gastrointestinal microbiota characteristics and different HS responses of the host, thus providing evidence and new directions for future studies aimed at understanding HS in individual calves (gut microbiota-host interactions).

## 1. Introduction

Increasing global temperatures adversely affect the welfare, productivity, and economic viability of dairy cattle [1,2]. Within a defined range of environmental temperatures, known as the thermoneutral zone, livestock are capable of maintaining a stable body temperature, thereby minimizing physiological expenditures and maximizing overall productivity [3]. Heat stress (HS) arises when livestock experience an elevation in body temperature beyond their capacity to adequately dissipate heat and maintain thermal equilibrium. This imbalance is often triggered by a combination of factors, including ambient temperatures exceeding the thermoneutral zone, high humidity levels, and limited air circulation [4,5]. This accumulation of heat can result in diminished productivity and reproductive efficiency, as well as higher mortality rates among livestock. In the United States, the livestock industry incurs annual economic losses ranging from 1.69 billion to 2.36 billion due to HS [6]. HS adversely impacts both the dry matter intake and growth performance of calves. Studies have indicated that calves born during summer months generally exhibit decreased average daily gain and reduced starter dry matter intake compared to those born in winter [7,8]. Furthermore, research has shown that calves exposed to HS conditions demonstrate a decline in growth indicators, including dry matter intake and average daily gain, in comparison to calves not subjected to such stress [9,10,11]. During HS, calves will gain more heat from the environment and metabolic processes than they will lose through radiation, convection, evaporation, and conduction [3]. As a result, heat accumulates in calves, leading to an increase in core temperature. Yazdi et al. [9] reported that rectal temperatures of calves under HS conditions were significantly higher than those of thermoneutral zone. Gastrointestinal microorganisms have a symbiotic relationship with the host, whereas in heat stress situations the microbial ecosystem is found to be disrupted (i.e., dysbiosis) [12,13,14]. Previous studies of HS have compared differences in calves under different temperature and humidity indices, but it is not clear whether there are differences in the degree of HS in calves under the same heat stress conditions. Sun et al. [15] showed differences in growth performance and blood metabolites among ducks with different behaviors by studying ducks exhibiting different active avoidance behaviors under the same stressor.

Therefore, the aim of this study was to determine the relationship between the degree of HS response in calves and gut flora by examining different HS responses in calves in the same thermal environment. We hypothesized that the gut microbiota is related to host metabolism and thus influences the different degree of HS response in calves.

## 2. Materials and Methods

The study procedures employed in this research were examined and authorized by the Institutional Animal Care and Use Committee at Inner Mongolia Agricultural University in China (Approval No. 20240017) in 23 December 2023, and adhered strictly to the fundamental principles guiding the welfare and utilization of laboratory animals.

### 2.1. Animal Management and Experiment Design

The research, initiated on 8 August 2024 and concluding on August 20th of the same year, took place at Duyang Dairy Co., Ltd., situated in Long County, Baoji, China (coordinates: 34.83° north latitude and 106.93° east longitude). All calves utilized in this study were sourced from this dairy farm, which is located in an area characterized by a temperate continental monsoon climate. Prior to the commencement of the experiment, each calf underwent a thorough clinical examination and was continuously monitored throughout the experimental duration. Each calf received pasteurized milk in its own open bucket, which was meticulously cleaned and allowed to drip dry after each feeding session in preparation for the next. Throughout the experimental period, the preweaning calves were fed a daily allotment of 9 L of pasteurized milk, split evenly into two meals at 06:00 and 16:00. For this experiment, a total of 30 preweaning calves were chosen. Their respiratory rates and rectal temperatures were recorded daily at 06:00 and 14:00. Additionally, the ambient temperature (T) and relative humidity (RH) were measured in the mornings (06:00) and afternoons (14:00) each day throughout the experimental period. These measurements were taken every 10 min using Hobo Pro Series Temp probes (manufactured by Onset Computer Co., Pocasset, MA, USA). The probes were positioned approximately 1.0 m above the ground. The temperature-humidity index (THI) was calculated [16]:THI = (1.8 × T +32) − (0.55 − 0.0055 × RH) × (1.8 × T − 26)

The experiment lasted 13 days and the THI results during the experiment period are shown in Table 1. Calves were ranked according to rectal temperature and respiratory rate detected at 14:00, respectively, and the most extreme 20 calves were selected and divided into two treatments: a high HS response group (H) and low HS response group (L) with 10 calves per group. For sample collection on day 11 of the experiment, after fasting for 4 h following morning feeding, individual blood samples were collected in sodium heparin tubes via jugular vein puncture. The blood samples underwent centrifugation at 1800× *g* for a duration of 10 min at a temperature of 4 °C. Subsequently, the serum was separated, transported to the laboratory, and kept at −80 °C for future analysis. Additionally, fecal samples and rumen fluid samples were gathered and preserved at −80 °C for subsequent examination. Using enzyme-linked immunoassay kits sourced from AngleGene BioTechnology Co., Ltd. in Nanjing, China, the serum samples were assayed for interleukin-2 (IL-2), interleukin-4 (IL-4), interleukin-6 (IL-6), tumor necrosis factor α (TNF-α), cortisol, serum amyloid A (SAA), haptoglobin (HP), heat shock protein 70 (HSP-70), glucose (GLU), insulin (INS), total fatty acids (TFA), and triiodothyronine (T3). These analyses were conducted on a microplate reader (Biotek Synergy H1, manufactured by BioTek Instruments, Inc., Winooski, VT, USA), following established protocols and adhering to the manufacturer’s instructions, as previously reported [17].

### 2.2. DNA Extraction, PCR Amplification, and 16S rRNA Sequencing

Genomic DNA was extracted from fecal samples using the hexadecyl trimethyl ammonium bromide (CTAB) method. Post-extraction, the quality assessment of the DNA samples was conducted on 1% agarose gels, while their concentration and purity were quantified utilizing a NanoDrop 2000 UV-vis spectrophotometer (Thermo Scientific, Waltham, MA, USA). To examine the microbial community composition, the hypervariable V3-V4 regions of the 16S rRNA genes were targeted for amplification, followed by sequencing. Sequencing was conducted by Shanghai Majorbio Bio-Pharm Technology Co., Ltd. (Shanghai, China), located in Shanghai, China, utilizing the Illumina MiSeq PE300 platform. The primer pair used consisted of 338F (5′-ACTCCTACGGGAGGCAGCA-3′) and 806R (5′-GGACTACHVGGGTWTCTAAT-3′). The PCR conditions adhered strictly to those outlined previously by Wu et al. [17]. The raw sequences of the 16S rRNA gene underwent preliminary processing utilizing the fastp tool, specifically version 0.20.0, which involved demultiplexing and stringent quality filtering. Following this, FLASH, version 1.2.7, was employed to merge these sequences. Further refinement was applied based on well-established criteria previously outlined by Wu et al. [17]. Operational taxonomic units (OTUs) were grouped by Uparse, applying a 97% similarity threshold for clustering. To guarantee the accuracy of the data, chimeric sequences were detected and excluded using Usearch (version 7.0), following the procedures outlined by Edgar [18]. The (Ribosomal Database Project) RDP Classifier algorithm was utilized to assign taxonomic classifications to each OTU by comparing their representative sequences against the Silva 16S rRNA database (specifically version 138). A threshold of 70% confidence was applied, in accordance with the methodology previously described by Wang et al. [19]. The gut microbiota was analyzed bioinformatically using the Majorbio Cloud platform, which is accessible via the web at https://cloud.majorbio.com (accessed on 27 March 2025). Utilizing Mothur version 1.30.1 and the curated OTU data, various alpha diversity metrics were computed, including the number of observed operational taxonomic units (OTUs), the Chao1 estimator of richness, the Shannon diversity index, and Good’s coverage. To evaluate the similarities among microbial communities across different samples, principal coordinate analysis (PCoA) was conducted based on Bray-Curtis dissimilarity, using the Vegan package in its version 2.5-3. LEfSe (Linear Discriminant Analysis Effect Size) [20] was utilized to detect bacterial taxa, ranging from phyla to genera, that exhibited significant abundance differences among the various groups. This analysis was conducted using an LDA score threshold of greater than 2 and a statistical significance level of *p* < 0.05.

### 2.3. Metabolomics Analysis of Serum Samples and Data Processing

For the serum metabolome analysis, a precise extraction protocol was followed [21]. In the context of system conditioning and quality assurance, a composite quality control (QC) sample was formulated by combining equal aliquots of all individual samples. This QC sample underwent processing and testing procedures identical to those applied to the analytical samples. Regular intervals (specifically, every 5 to 15 samples) witnessed the injection of the QC sample, serving to oversee the stability and consistency of the analytical process. For comprehensive procedural details, please refer to prior studies [21]. The LC-MS/MS analysis of sample was performed using a Thermo UHPLC-Q Exactive system, equipped with an ACQUITY HSS T3 column (100 mm × 2.1 mm internal diameter, 1.8 μm; Waters, USA) at Majorbio Bio-Pharm Technology Co., Ltd. (Shanghai, China). The raw data obtained from LC/MS was pretreated using Progenesis QI software (Waters Corporation, Milford, CT, USA), resulting in the export of a three-dimensional data matrix in CSV format. This preprocessing resulted in the creation of a three-dimensional data matrix in CSV format. This matrix contained crucial information, such as sample specifics, metabolite identifiers, and the intensities of mass spectral peaks. Prior to analysis, internal standard peaks and identified false positives, which included noise, column bleed artifacts, and peaks stemming from derivatized reagents, were removed from the data matrix. This matrix was then dereplicated and underwent peak aggregation. Simultaneously, metabolite identification was carried out by consulting multiple databases, notably the Human Metabolome Database (accessible via http://www.hmdb.ca/, accessed on 27 March 2025), Metlin (hosted at https://metlin.scripps.edu/, accessed on 27 March 2025), and the Majorbio Database. The collected data underwent analysis on the free online majorbio cloud platform (available at cloud.majorbio.com). Metabolic features that were consistently detected in at least 80% of any given sample set were selected for further evaluation. Following this filtering step, for the samples where metabolite concentrations fell below the quantitation limit, the minimum metabolite values were assigned. Normalization of the metabolic features was achieved by summing them up. To mitigate errors stemming from sample handling and instrumental variability, the intensity of mass spectral peaks in the samples was normalized using the sum normalization technique, yielding a normalized data matrix. Additionally, variables displaying a relative standard deviation (RSD) exceeding 30% in the quality control (QC) samples were omitted. Subsequently, a log10 transformation was applied to generate the definitive data matrix for further analysis. Orthogonal partial least squares discriminant analysis (OPLS-DA) was employed for comparisons between the L and H groups, augmented by a Student’s *t*-test for statistical validation. Metabolites exhibiting significant differences in abundance between the two groups were determined based on the criteria that their variable importance in projection (VIP) values surpassed 1.0 and their *p* < 0.05. The significantly differentially abundant metabolites were further analyzed by summarizing and mapping them to their respective biochemical pathways through metabolic enrichment and pathway analysis, utilizing a database search (KEGG). These metabolites could be classified based on the pathways they engage in or the functions they fulfill. To determine if a functional node contained a specific group of metabolites, enrichment analysis was employed, shifting the focus from annotating individual metabolites to annotating metabolite groups. The impact of the metabolic pathways and metabolite set enrichment was assessed using the “stats” package in R and the “scipy” package in Python, respectively. KEGG metabolic pathways were deemed significantly enriched if they fulfilled a *p* < 0.05.

### 2.4. Statistical Analysis

Prior to initiating the analysis, we conducted an exhaustive examination of all data utilizing SPSS Statistics software (IBM Corp., Armonk, NY, USA; version 24.0) to ascertain both the normality and homogeneity of variances. When the data deviated from normality assumptions, logarithmic transformations [specifically, base-10 logarithms, log10(x)] were employed to ensure a more uniform distribution of the data. For the present study, each calf was treated as a distinct experimental unit, amounting to a total of ten units (n = 10). The Student’s *t*-test was utilized within SPSS Statistics to analyze serum indices. To assess the likelihood of co-occurrence among microbial entities or between microorganisms and metabolites, we conducted Spearman correlation analysis (with a correlation coefficient threshold of r > 0.5 and a statistical significance of *p* < 0.05) utilizing the Networkx tool sourced from Shanghai Majorbio Bio-Pharm Technology Co., Ltd. (Shanghai, China). Additionally, we explored whether the assessment of biomarkers related to the gastrointestinal microbiota in dairy calves could enhance our predictive capabilities for high and low HS responses [22]: The process involved two main steps: (1) dividing the data into two distinct groups based on HS response, and (2) evaluating the area under the receiver operating characteristic curves (AUROC) using GraphPad Prism software (version 8.0.2, GraphPad Software, Inc., San Diego, CA, USA). All figures were constructed using GraphPad Prism, and statistical significance was determined at a *p* < 0.05.

## 3. Results

Table 1 depicts the THI data collected during the experimental period. At 14:00, THI had a minimum value of 76.62 and a maximum of 82.76. Conversely, at 6:00, THI ranged from a minimum of 64.19 to a maximum of 71.45. THI at 14:00 was significantly higher than at 06:00 (*p* < 0.05, Table 1).

There were no differences in weaning weight and daily gain between the two groups of calves during the trial (*p* > 0.05, Table 2).

During the HS period, there were significant differences in rectal temperature and respiratory rate between the two groups of calves (Table 2; *p* < 0.05). However, during the non-HS period, there were no differences in respiratory rate and rectal temperature among the calves (*p* > 0.05, Table 3).

### 3.1. Blood Indicators Analyses

The blood indicators are shown in Figure 1. Compared to group L, calves in group H had higher concentrations of IL-2 (*p* = 0.03), HSP-70 (*p* = 0.02), and TFA (*p* = 0.05) in their serum, while there were no differences in other serum indicators (*p* > 0.05) (Figure 1).

### 3.2. Microbiome Analyses

We analyzed the bacterial communities present in both the rumen and feces to explore the influence of alterations in microbial populations on different levels of HS responses. We obtained a total of 1,241,144 clean reads from rumen fluid samples (Appendix A). The Good’s coverage rate (≥99.5%, as shown in Appendix A) ensures that the sequencing depth we chose is sufficient to comprehensively cover the microbial diversity of each sample. In addition, there was no significant difference in sequencing depth between the two groups of rumen fluid samples. Upon comparison, it was found that group H possessed a higher ASV (Amplicon Sequence Variant) count than group L (Figure 2A). Statistical data on α-diversity and β-diversity indicate no differences in the uniformity and richness of rumen microorganisms at the ASV level (*p* > 0.05; Appendix A). The species abundance at various taxonomic levels, including phylum and genus, was assessed and ordered. At the phylum level, the rumen was predominantly populated by *Firmicutes* and *Bacteroidetes* (Figure 2B). At the genus level, the five most abundant genera were *Prevotella*, *Lachnospiraceae_NK3A20_group*, *norank_o__Clostridia_UCG-014*, *Ruminococcus*, and *norank_f__Muribaculaceae* (Figure 2C). Linear discriminant analysis (LDA) (Figure 2D) uncovered the presence of one bacterial order, four bacterial families, and eight bacterial genera. No significant statistical differences were observed in the relative abundances of these species at the phylum taxonomic level. However, at the genus level, eight distinct bacterial genera, among the top ten in relative abundance (Figure 2E), were identified. Based on the outcomes of Latent Dirichlet Allocation (LDA), eight bacterial genera were identified as the pivotal and efficacious differentiators between the L and H groups. These included *g__Prevotella*, *g__Ruminococcus*, *g__Pseudoramibacter*, *g__Intestinimonas*, *g__Clostridium sensu stricto 1*, *g__Allorhizobium-Neorhizobium-Pararhizobium-Rhizobium*, *g__norank_f__Weeksellaceae*, and *g__Jeotgalibaca*. These eight genera, curated from the LDA results, were chosen as biomarkers for the rumen microbiota. Further correlation analysis revealed a negative correlation between *g_Ruminococcus* and the respiratory rate of calves (*p* = 0.04; r = −0.45; Figure 2F). By utilizing the shared top 52 bacterial genera (each with a relative abundance exceeding 0.1%) from both groups, a co-occurrence network analysis was conducted. This analysis revealed a comprehensive picture of interactions, encompassing a total of 236 co-occurrence associations. Specifically, within the L group (depicted in Figure 2H), there were 53 negative relationships and 83 positive relationships. Conversely, in the H group (illustrated in Figure 2I), there were 9 negative relationships and a significantly higher number of positive relationships, totaling 115. Notably, 24 of these relationships were consistent between the two groups (Figure 2G). The findings suggest a shift in the symbiotic dynamics of the rumen microbiota between the two groups, characterized by a greater degree of cooperation within the H group and a relatively higher level of competition within the L group. Appendix A provides further detailed information regarding these findings.

We acquired a total of 1,325,482 clean reads from the rectal fecal samples (see Appendix A). The Good’s coverage index, which reached ≥ 99.5% (as indicated in Appendix A), confirms that our chosen sequencing depth was adequate for comprehensively capturing the microbial diversity present in each sample. In addition, there was no significant difference in sequencing depth between the two groups of fecal samples. Upon comparison, it was found that group L possessed a higher ASV count than group H (Figure 3A). Analysis of α-diversity and β-diversity statistics revealed no significant differences in the evenness and abundance of rumen microorganisms at the ASV level (*p* > 0.05; refer to Appendix A for details). The species abundance at various taxonomic levels, including phylum and genus, was assessed and ordered. At the phylum level, the fecal microbiota was predominantly characterized by Firmicutes and Bacteroidetes (as shown in Figure 3B); at the genus level, the top five most abundant genera were *norank_o__Clostridia_UCG-014*, *Blautia*, *norank_f__Muribaculaceae*, *Peptoclostridium*, *UCG-005*, and *Lactobacillus* (Figure 3C). Linear discriminant analysis (Figure 3D) revealed the presence of a solitary bacterial class and order, one bacterial family, and seven bacterial genera. No significant statistical variations were observed in the proportional abundances of species at the phylum level. However, seven distinct bacterial genera, among the top 10 in relative abundance, were identified at the genus level (Figure 3E). Following linear discriminant analysis, six genera—*g__Olsenella*, *g__Family_XIII_AD3011_group*, *g__Catenisphaera, g__Pseudoramibacter*, *g__norank_f__Oscillospiraceae*, and *g__Colidextribacter*—were selected as the primary effective bacteria differentiating the L and H groups. From the conducted analysis, eight specific genera were identified and designated as biomarkers for the fecal microbiota. By examining the top 52 bacterial genera from both groups, which possessed a relative abundance exceeding 0.1%, a co-occurrence network analysis unveiled a comprehensive total of 287 associative interactions. Specifically, the L group exhibited 66 negative and 109 positive relationships (Figure 3H), whereas the H group showed 31 negative and 116 positive relationships (Figure 3I). Thirty-five of these relationships were common to both groups (Figure 3G). These findings suggest a shift in the symbiotic relationships within the fecal microbiota of the two groups.

To explore the relationship between ruminal and intestinal microbiota within each group, a correlation analysis was performed. This involved the use of a Two-Matrix Correlation Heatmap, which was based on the Pearson correlation coefficient, to evaluate the alpha diversity index of the microbiota. The results of this analysis are illustrated in Figure 4. The results revealed that calves in the L group demonstrated negative correlations across all associations between their ruminal and intestinal microbiota, as shown in Figure 4A. For example, a strong positive correlation (r = 0.828; *p* = 0.003) was noted between the ruminal Chao index and the intestinal Chao index. Furthermore, an intriguing finding in the H group, as depicted in Figure 4B, revealed that the correlations were predominantly centered around indices of community richness. In contrast, they were less focused on diversity-related indices, such as the Shannon and Simpson indices, which are commonly used as indicators of community diversity and reflect both the abundance and evenness of species within an ecosystem. Nevertheless, no statistically significant correlations were identified between the ruminal and intestinal microbiota in the H group, as illustrated in Figure 4B (*p* > 0.05). Appendix A provides further details on this matter.

### 3.3. Differential Metabolites Screening and Analysis of Host Serum

For both the L and H groups, we conducted a metabolomic examination of small metabolites present in blood serum. To distinguish between the metabolite profiles of the two groups, we produced OPLS-DA score plots for both positive and negative ionization modes. The results of the multivariate analysis are depicted in Figure 5A and Figure 5B, respectively. It is noteworthy that all serum samples collected from the calves are encompassed within the 95% confidence ellipse depicted in the score plots. To ascertain the dependability of our OPLS-DA model, we examined the R2Y and Q2 parameters. In detail, for the positive ion mode, the serum samples exhibited an R2Y of 0.873; contrastingly, in negative ion mode, the R2Y amounted to 0.76. For a more comprehensive evaluation of our models’ robustness and predictive accuracy, we utilized a seven-fold cross-validation method augmented with permutations, and after 200 permutation cycles, we computed the Q2 intercept values. The results of this analysis are depicted in Appendix A. The permutation testing procedure generated Q2 intercept values of −0.041 for positive ion mode serum samples and −0.047 for negative ion mode samples, respectively. Remarkably, both positive and negative datasets exhibited clear separation between the L and H groups, validating the efficacy of the OPLS-DA model in distinguishing these two groups. Our analysis encompassed a total of 1479 metabolites in the serum of calves, with 330 detected in positive ion mode and 808 in negative ion mode. By applying the Student’s *t*-test (*p* < 0.05) and setting a VIP score threshold above 1.0 derived from the OPLS-DA model, we pinpointed metabolites that demonstrated significant variations between the two groups among all detected metabolites. To visually represent these notable discrepancies, we utilized volcano plots (depicted in Figure 5C,D for positive and negative ion modes, respectively). These plots distinctly underscore substantial changes in a multitude of plasma metabolites across the groups, with particular emphasis on 17 metabolites that exhibited significant differences (out of 122 in positive ion mode and 53 in negative ion mode). A comprehensive enumeration of the metabolites showing significant variations is provided in Appendix A. This enumeration encompasses alkaloids and their derivatives, benzenoids, lipids and lipid-like molecules, nucleosides, nucleotides, and their counterparts, organic acids and their derivatives, organic compounds containing oxygen, organoheterocyclic compounds, as well as phenylpropanoids and polyketides (also refer to Appendix A for further details). Regarding the metabolic pathways analyzed through KEGG topology, the pathways exhibiting significant differences (*p* < 0.05) and notable impacts (with impact values exceeding 0) were pyruvate metabolism, lysine biosynthesis, lysine degradation, and the citrate cycle (TCA cycle). Notably, all these pathways were enriched within the L group, as depicted in Figure 5E. The metabolites abundant in these distinct metabolic pathways included (S)-5-Amino-3-oxohexanoate, 2,6-Diaminopimelic acid, allysine, malic acid, and fumaric acid (as listed in Appendix A). These were subsequently selected as biomarkers for the small-molecule metabolites within the host’s metabolome.

### 3.4. Microbe–Host Metabolite Interactions Associated with Cattle Heat Stress Response

To delve deeper into the complex relationships among specific entities, we performed an additional topological network analysis, utilizing eight biomarker genera identified within the rumen microbiota. This analysis revealed two distinct phenotypes, thirteen serum indicators, and five small-molecule metabolites functioning as biomarkers in the serum metabolome. Subsequently, nodes were classified as hubs or non-hubs based on their degree, closeness centrality, and betweenness centrality metrics, as derived from the topological analysis of the network. Our results prominently showed that HSP-70 (a serum parameter), malic acid and fumaric acid (as serum metabolites), along with *g__Ruminococcus* and *g__Clostridium_sensu_stricto_1* from the rumen microbial genera, and *g__Pseudoramibacter* from the fecal microbial genera, emerged as key nodes in the analysis (depicted in Figure 6 and Appendix A). These hubs were found to be correlated with each other (r > 0.5; *p* < 0.05). Additionally, we assessed the predictive power of these hubs in relation to the host’s heat stress response by employing an AUROC curve analysis (see Appendix A for details). Significantly, our analysis revealed thatHSP-70, malic acid, *g__Ruminococcus* (derived from the rumen microbiota), and *g__Pseudoramibacter* (originating from the fecal microbiota) could accurately differentiate calves based on HS responses (with an AUROC value exceeding 0.70 and a *p* < 0.05; as illustrated in Figure 7).

## 4. Discussion

There are many studies on the effects of HS on calves, including the effects of HS on growth performance [10], inflammation [12], gastrointestinal microbiology [12,23,24] in preweaning calves, and on productive performance in later adulthood; in addition, maternal HS during the second trimester results in limited growth and development of offspring calves, and maternal HS during the second trimester negatively affects the growth of second generation calves [9,25,26,27]. The results of the above studies have shown that HS can have serious negative effects on calves, such as reduced survival rates and increased incidence of disease [3]. But most of the current research on heat stress has focused on studies comparing differences in calves with and without HS [3,15,28,29]. It has been demonstrated that there are significant differences in the phenotype and blood metabolite composition of laying ducks under the same stressor [15]. Based on these findings, we hypothesized that calves do not respond equally to HS under identical heat stress conditions, mainly in terms of differences in gastrointestinal microbial ecosystems and blood metabolite composition.

In this study, we grouped calves based on respiratory rate and found 10 calves each with the highest and lowest respiratory rate according to their respiratory rate, which was significantly different between these two groups. Wang et al. [3] reported that an increase in respiratory rate precedes an increase in core body temperature, suggesting that it is feasible to use respiratory rate to judge calf HS. It has been shown that the rectal temperature of calves increased under HS conditions [30,31] and there was a significant difference in rectal temperature between the two groups under the conditions of the present study, which justifies the grouping in the present study.

Heat shock proteins (HSPs) represent a category of molecular “chaperones”—essentially assistants or facilitators—that are indispensable in sustaining cellular homeostasis and enhancing the ability of cells to endure high temperatures, thereby fostering thermotolerance [32]. One of the most widely studied proteins is HSP70, which is known to be induced by HS, and its expression level is increased in various tissues of cows under HS conditions [33]. Rakib et al. [34] reported that HSP70 can be used as a potential indicator of heat stress in dairy calves, and in the present study significant differences in HSP70 were found between high and low HS response groups. IL-2 is often used as a parameter to evaluate the state of the body’s immune function and is a pro-inflammatory factor [35]. In another study of HS, it was found that HS resulted in increased inflammation in calves [12]. In the current study, calves in the H HS-responsive group exhibited significantly elevated IL-2 concentrations compared to those in the L HS-responsive group, indicating a more pronounced inflammatory response in the high HS-responsive calves. One of the hallmarks of heat stress acclimatization in dairy calves is a limited ability to mobilize body fat [36]. Significantly higher concentrations of TFA were found in the high HS group in the present study, whereas in other studies it was similarly found that HS leads to higher concentrations of FA [9,37]. This implies that calves with varying HS responses also differ in their metabolic capabilities. Our study has identified biomarkers within blood parameters, specifically HSP70, which may potentially predict the varying levels of HS response.

Further, we analyzed differences in rumen and fecal microbial composition. First, rumen microbial diversity (α and β diversity) did not differ between the two groups, which may be caused by the fact that the rumen is not fully developed in calves at this stage [38]. No effect of HS on rumen [39] and gut microbial [12] diversity was found in previous studies, which is consistent with the results of the present study. The results of the above experiments show that HS does not change the diversity of microorganisms, but it does change the composition of microorganisms. In addition, the proportion of beneficial and harmful microorganisms in the rumen of calves changed in response to different HS. For example, *Prevotella* produces short-chain fatty acids and is able to prevent gastrointestinal infections by competing with pathogenic bacteria for binding sites on epithelial cells [40], and this bacterium was the dominant genus in the rumen in the present study, which is consistent with reports by others [41,42]. In the present study, the abundance of *Prevotella* was found to be significantly lower in calves from the high HS response group than in calves from the low HS response group. *Rumenococcus* is one of the most efficient bacteria for breaking down carbohydrates [43,44] and has the ability to stabilize the intestinal barrier and reverse diarrhea [45]. *Ruminococcus* abundance was found to be significantly decreased in the high HS response group in this study, and in another study it was reported that *Ruminococcus* was significantly enriched in healthy calves [46]. In one study, a strong correlation was found between changes in body temperature and microorganisms [47]. Meanwhile, a negative correlation between the abundance of Ruminococcus in the rumen and respiration rate was found in this study, which explains to some extent the inconsistency in respiration rate between the two groups. In this study, we identified biomarkers in rumen microorganisms, including *Ruminococcus* that can serve as potential predictors of different degrees of HS response. *Olsenella* was found to favor immune homeostasis in previous studies [48,49], and in the present study fecal *Olsenella* abundance was significantly higher in calves of the low HS response group than in the high HS response group, which suggests that calves of the high HS response group were less immune-competent. Our study revealed a negative correlation between the rumen and gut microbiota in the group of calves exhibiting a low HS response (e.g., negative correlation between the Chao index of the rumen microorganisms and the Chao index of the gut microorganisms), and a negative correlation between the rumen microbiota and gut microbiota of the control group with no treatments was found in our previous study [50], which was disrupted in the high HS response group, which may be a consequence of the difference between the two reasons for the differences in HS response across calves.

Metabolomics can provide insights into the roles of small-molecule metabolites, which are crucial in numerous biological processes [51]. In this study, we identified five metabolites ((S)-5-amino-3-oxohexanoate, 2,6-diaminopimelic acid, allysine, malic acid, and fumaric acid) involved in four major metabolic pathways (pyruvate metabolism, lysine biosynthesis, lysine degradation, and the citrate cycle (TCA cycle)). Produced as the final outcome of glycolysis, pyruvate originates from various cellular cytoplasm sources and is subsequently transported to the mitochondria, serving as the primary substrate for driving the carbon flow in the TCA cycle [52]. The TCA cycle is a major energy-generating pathway that occurs primarily in the mitochondrial matrix and is a central metabolic node linking carbohydrate, lipid, and amino acid metabolism [53]. This suggests that different HS responses in calves may be related to energy metabolism, which needs to be verified by further experiments. The abundance of pyruvate metabolism and TCA cycle-related metabolites malic acid and fumaric acid was down-regulated in calves from the high HS response group compared to calves from the low HS response group. Malic acid reduces inflammation and improves amino acid metabolism and energy metabolism [54]. Some studies have shown that fumaric acid is a growth promoter, enhances the body’s immunity, reduces inflammation, and improves intestinal health [55,56]. This verifies at the metabolic level that inflammation does vary among HS-responsive calves. These intriguing findings not only reveal the varying degrees of HS responses but also underscore the value of utilizing metabolomics in such analyses. This study has identified small-molecule metabolites associated with these vital metabolic pathways, which are involved in these biological processes. Specifically, we have identified malic acid and fumaric acid, among other serum metabolome biomarkers, as potential candidates for predicting various intensities of HS responses. These findings support our results, suggesting that calf responses to HS are nevertheless variable, both phenotypically and within the organism.

## 5. Conclusions

In conclusion, rectal temperature, rumen *Ruminococcus*, serum HSP70, malic acid, and fumaric acid can serve as distinguishable markers of calves’ different HS response. This study reveals the relationship between gastrointestinal microorganisms, metabolites, and different HS responses of the host, and provides a new research approach for future calf HS. Individual calves respond differently to heat stress, and the future management of heat stress interventions in ruminants will require a shift from group to individual HS intervention management.

## Figures and Tables

**Figure 1 microorganisms-13-00801-f001:**
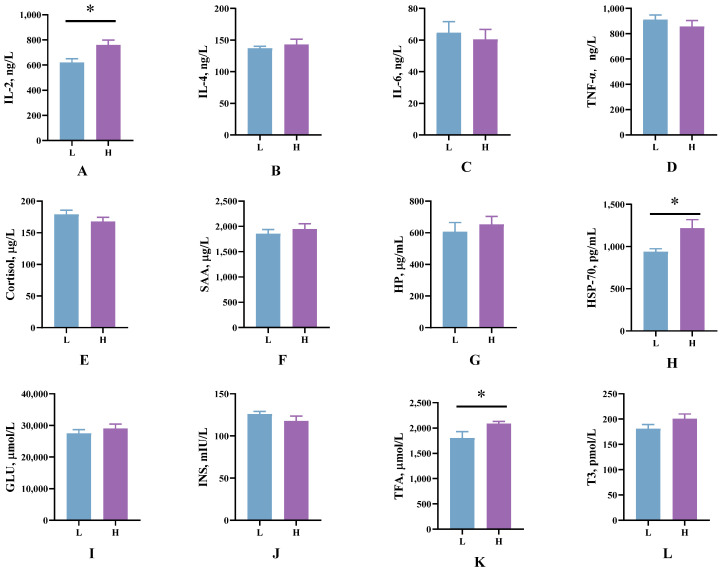
Comparison of plasma and serum biochemical analysis. Serum levels of (**A**) IL-2, (**B**) IL-4, (**C**) IL-6, (**D**) TNFα, (**E**) Cortisol, (**F**) SAA, (**G**) HP, (**H**) HSP-70, (**L**) T3; plasma levels of (**I**) GLU, (**J**) INS, and (**K**) TFA among high HS response (**H**) and low HS response (**L**) groups. * indicates significance (*p* < 0.05). n = 10.

**Figure 2 microorganisms-13-00801-f002:**
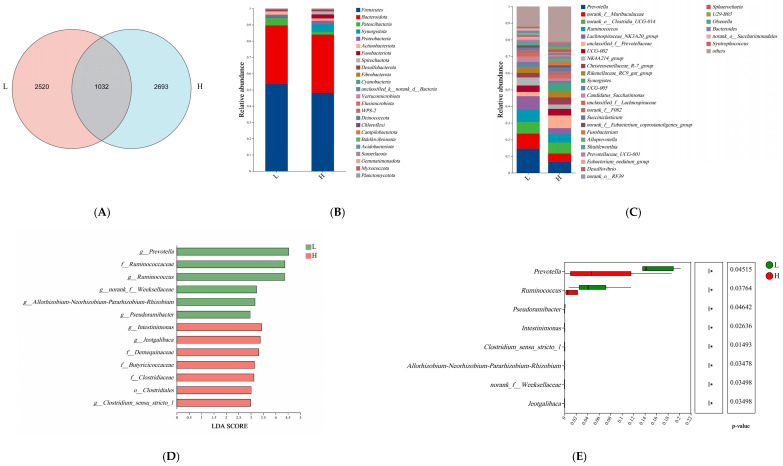
A comparative taxonomic analysis was conducted on the rumen microbiota of calves, categorizing them into two groups based on their heat stress (HS) response: high (H group) and low (L group). Panel (**A**) presents a Venn plot at the genus level, showing the microbial overlap between the groups. The composition of the rumen microbiota at the phylum (**B**) and genus (**C**) levels is also detailed. To accurately identify distinct species within the rumen microbiota, we employed both the LEfSe bar chart (**D**) and Student’s *t*-test (**E**), spanning from the phylum to genus levels. Additionally, a heatmap (**F**) was generated to depict the relationship between microbial genera in the rumen and blood parameters. A Venn plot (**G**) visualizes the common interactions between these groups, with red edges indicating positive relationships and blue edges signifying negative relationships. L group (**H**): low HS response, H group (**I**): high HS response, the size of each node in the plot corresponds to the mean abundance of the respective microbial species. Differences were defined as significance with * *p* < 0.05, ** *p* < 0.01, n = 10.

**Figure 3 microorganisms-13-00801-f003:**
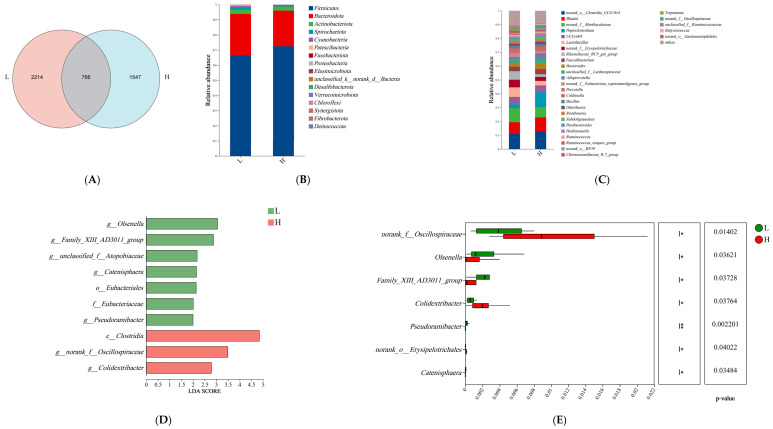
A comparative taxonomic analysis was performed on the intestinal microbiota of calves, categorizing them into two groups based on their heat stress (HS) response: high (H group) and low (L group). Panel (**A**) displays a Venn plot at the genus level, illustrating the microbial overlap between the two groups. The composition of the intestinal microbiota at the phylum (**B**) and genus (**C**) levels was also examined. To discern differential species within the intestinal microbiota, spanning from phylum to genus levels, two methods were utilized: the LEfSe bar chart (**D**) and Student’s *t*-test (**E**). Additionally, a heatmap (**F**) was constructed to visualize the correlations between microbial genera in the intestine and blood indicators. In this study, the H group comprised calves exhibiting a high heat stress (HS) response, whereas the L group consisted of calves with a low HS response. A Venn plot (**G**) was used to depict the common interactions between these groups, with red edges signifying positive relationships. L group (**H**): low HS response, H group (**I**): high HS response, the size of each node in the plot corresponds to the mean abundance of the respective microbial species. Differences were defined as significance with * *p* < 0.05, ** *p* < 0.01, n = 10.

**Figure 4 microorganisms-13-00801-f004:**
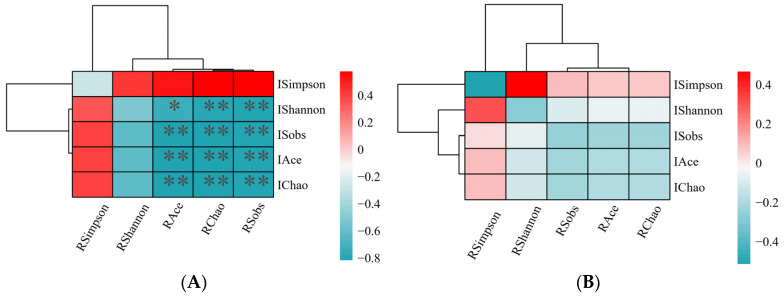
A correlation analysis was performed to investigate the association between ruminal and intestinal microbiota within each group, utilizing a Two-Matrix Correlation Heatmap based on the Pearson correlation to assess the alpha diversity index of the microbiota. The outcomes of this analysis are displayed distinctly for the L group (representing low heat stress response) and the H group (representing high heat stress response) in figures (**A**) and (**B**), respectively. In these heatmaps, the prefix R denotes ruminal outcomes, while I represents intestinal outcomes. Significance was established at * *p* < 0.05, ** *p* < 0.01. n = 10.

**Figure 5 microorganisms-13-00801-f005:**
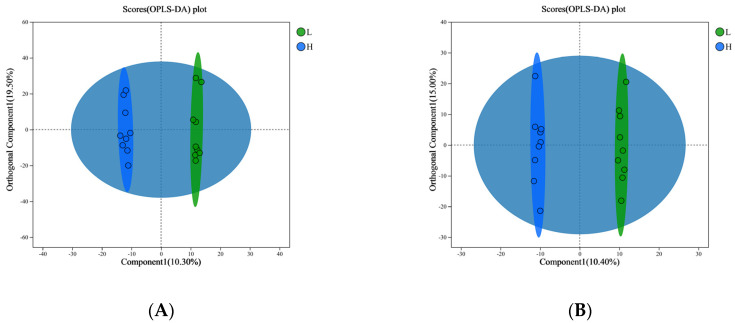
The aim of this research was to identify differences in small-molecule metabolites and emphasize enriched metabolic pathways within the serum metabolome of calves that displayed low (L) compared to high (H) heat stress responses. Orthogonal partial least squares-discriminant analysis (OPLS-DA) was utilized to produce score plots for both positive (**A**) and negative (**B**) ion modes. For the purpose of visualizing the metabolites that displayed significant differences, volcano plots were generated for both positive (**C**) and negative (**D**) ion modes. Furthermore, an analysis was conducted to enrich the differential metabolic pathways within the calves’ endogenous metabolome (**E**), where pathways of significance and influence were identified using a *p* < 0.05 and KEGG topology analysis (impact value > 0). n = 10.

**Figure 6 microorganisms-13-00801-f006:**
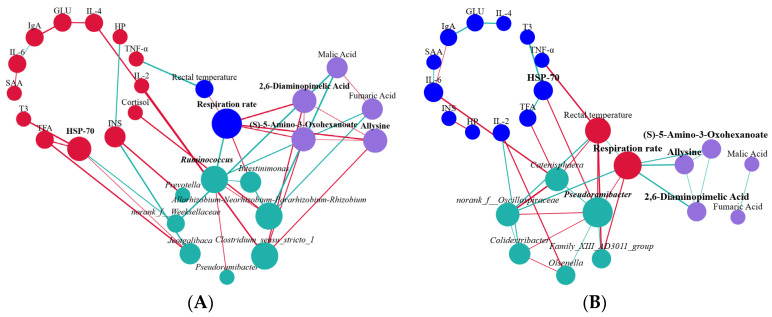
Using network topological analysis, we aimed to clarify the interactions among phenotype, rumen bacteria, blood indices, and serum macromolecule metabolites. (**A**): Rumen; (**B**): Fecal. The serum metabolome encompasses pathways related to pyruvate metabolism and the TCA cycle (including malic acid and fumaric acid), lysine biosynthesis (involving 2,6-diaminopimelic acid and allysine), and lysine degradation ((S)-5-amino-3-oxohexanoate and allysine). Green edges in the network represent negative relationships, whereas red edges signify positive relationships. The size of the nodes signifies their degree, closeness centrality, and betweenness centrality, with emphasis given to those designated as hubs within the respective networks.

**Figure 7 microorganisms-13-00801-f007:**
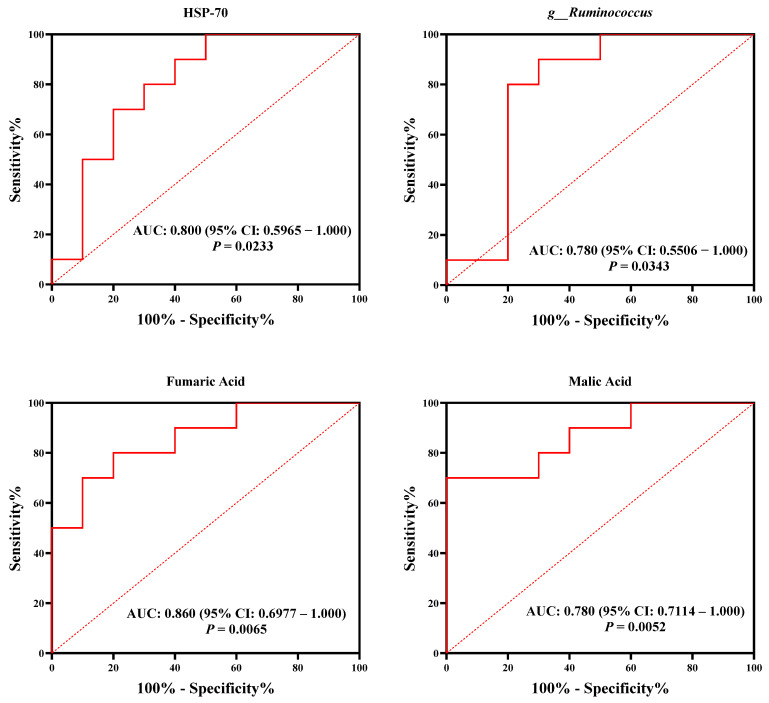
Employing serum metabolites, ruminal, and intestinal bacteria to anticipate host HS response, as interpreted by the area under the receiver operating characteristic curve (AUROC). The confidence interval (CI) was used to measure the precision of our predictions. Statistical significance was determined at *p* < 0.05.

**Table 1 microorganisms-13-00801-t001:** Daily 14:00 and 06:00 ambient temperature (°C), relative humidity (%), and THI during the environmental HS exposure period from 8–20 August 2024, at the experimental calf farm in Shanxi.

	6:00	14:00	SEM	*p*-Value
Temperature, °C	20.01	29.77	1.36	<0.0001
Relative humidity, %	87.41	58.08	13.38	<0.0001
THI	67.43	79.08	1.74	<0.0001

**Table 2 microorganisms-13-00801-t002:** Body weight (BW) and average daily gain (ADG) of calves from high HS response (H) and low HS response groups.

	L	H	SEM	*p*-Value
Initial BW, kg	39.70	38.80	4.42	0.65
Final BW, kg	100.87	103.37	2.59	0.56
ADG, kg/d	0.87	0.92	0.03	0.35

L: low heat stress response, H: high heat stress response; SEM denotes standard error of the mean; n = 10.

**Table 3 microorganisms-13-00801-t003:** Respiration rate and rectal temperature of calves from high HS response (H) and low HS response groups.

	L	H	SEM	*p*-Value
6:00				
Respiration rate, breath/min	56.11	58.51	2.92	0.568
Rectal temperature, °C	39.01	39.08	0.04	0.164
14:00				
Respiration rate, breath/min	62.68	79.78	2.90	0.001
Rectal temperature, °C	39.10	39.36	0.04	<0.0001

L: low heat stress response, H: high heat stress response; SEM denotes standard error of the mean; n = 10.

## Data Availability

The 16S rRNA amplicon sequencing data produced during this research are publicly accessible in the NCBI database, cataloged under the BioProject ID PRJNA1217494, MetaboLights database—MTBLS9735.

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
