# Peer review of "Gastrointestinal Microorganisms and Blood Metabolites in Holstein Calves with Different Heat Stress Responses in the Same Hot Environment"

_microorganisms, 2025, doi:10.3390/microorganisms13040801_

Round 1

Reviewer 1 Report

Comments and Suggestions for Authors
  • The graphs generated using Python are excellent and highly illustrative.
  • The authors have been meticulous in their statistical analysis, verifying model assumptions (normality and homogeneity of variances) and applying logarithmic transformations when these assumptions were not met.
  • For variables measured on ordinal scales, a Spearman correlation analysis was conducted, which is appropriate.
  • Writing Corrections:
    • Page 2: "Materials and Menthods" should be corrected to "Materials and Methods."
    • Page 3. In line two, a reference is made to figure 1; however, no such figure exists. The correct reference should be table 1."
    • Page 5. In the results section, the second paragraph refers to rectal temperature and respiration rate characteristics, which should be located in table 2. However, the cited table contains information on initial weight, final weight, and average daily gain. Therefore, there is a discrepancy.
    • Page 6. There is a discrepancy between the description of table 3 and the information presented in the table regarding the variables described.
    • Correct the numbering of figure 3 to figure 1.
        • "Define the acronym ASV in the microbiome analyses section."
        •  Page 10. In line four, define the variable 'n' or, alternatively, correct the word.
        • Lines 106-108 (page 17): The statement regarding the effects of HS in pregnant cows negatively impacting calf growth and development, and that these effects may persist for two generations, should be carefully reworded. As currently written, it could be misinterpreted as the inheritance of acquired negative behaviors.

      Overall, the article makes a significant contribution to the understanding of HS in Holstein calves and its relationship with gastrointestinal microbiota and blood metabolites. With minor revisions, this work will be a valuable addition to the scientific literature.

Author Response

Comments 1: Page 2: "Materials and Menthods" should be corrected to "Materials and Methods."

Response 1: Thank you very much for taking the time to review this manuscript. Revisions have been made (Lines 67).

Comments 2: Page 3. In line two, a reference is made to figure 1; however, no such figure exists. The correct reference should be table 1."

Response 2: We are very sorry for our incorrect. Revisions have been made (Lines 240 and 245). And we have made changes throughout the text.

Comments 3: Page 5. In the results section, the second paragraph refers to rectal temperature and respiration rate characteristics, which should be located in table 2. However, the cited table contains information on initial weight, final weight, and average daily gain. Therefore, there is a discrepancy.

Response 3: We are very sorry for our incorrect. Revisions have been made (Lines 230-233 and 224-225).

Comments 4: Page 6. There is a discrepancy between the description of table 3 and the information presented in the table regarding the variables described.

Response 4: We are very sorry for our incorrect. Revisions have been made (Lines 230-233).

Comments 5: Correct the numbering of figure 3 to figure 1.

Response 5: We are very sorry for our incorrect. Revisions have been made (all figures).

Comments 6: "Define the acronym ASV in the microbiome analyses section."

Response 6: We have made correction according the Reviewer's comments. Revisions have been made (Lines 256-257).

Comments 7: Page 10. In line four, define the variable 'n' or, alternatively, correct the word.

Response 7: We are very sorry for our incorrect. Revisions have been made (Lines 295).

Comments 8: Lines 106-108 (page 17): The statement regarding the effects of HS in pregnant cows negatively impacting calf growth and development, and that these effects may persist for two generations, should be carefully reworded. As currently written, it could be misinterpreted as the inheritance of acquired negative behaviors.

Response 8: We have made correction according the Reviewer's comments. Revisions have been made (Lines 446-449). 

in addition, maternal HS during the second trimester results in limited growth and development of offspring calves, and maternal HS during the second trimester negatively affects the growth of second generation calves.

Reviewer 2 Report

Comments and Suggestions for Authors

The manuscript aimed to evaluate differences in fecal microbiota and blood metabolomes in calves with different HS responses in the same warm environment. Although the manuscript is interesting, some points marked in the attached file must be addressed, since the lack of line numbering makes it difficult to indicate points for improvement or questions.

Authors must review the maximum number of words in the abstract according to the author guidelines since they are exceeded.

Figure 1 and 2 are not shown in the uploaded file.

The description of the groups formed should be clearer as it is somewhat confusing.

Author Response

Comments 1: Authors must review the maximum number of words in the abstract according to the author guidelines since they are exceeded.

Response 1: We have made correction according to the Reviewer's comments. The abstract word count has now been reduced to 198.

Comments 2: Figure 1 and 2 are not shown in the uploaded file.

Response 2: We are very sorry for our incorrect. We have made changes (all figures).

Comments 3: The description of the groups formed should be clearer as it is somewhat confusing.

Response 3: We have made correction according to the Reviewer's comments. A total of 30 test cows were selected for this trial, and rectal temperature and respiratory rate were measured for 13 consecutive days. Based on the rectal temperature, we ranked the 10 calves with the highest rectal temperatures at 14:00 as the high heat stress group, the 10 calves with the lowest rectal temperatures as the low heat stress group, and the 10 calves with rectal temperatures in the middle of the range were excluded from the trial.

Reviewer 3 Report

Comments and Suggestions for Authors

Please define the objective of the study in your introduction section. 

just 13 days is enough to select cows with different THI answers. Authors may please justify it?

There is no Figure 1 and Figure 2. Figure 1 yet is cited on page 3.

There are many typing errors that show the poor preparation of the manuscript, commas where there should be periods, spacing errors, etc. (Some pages do not have line numbers, which makes it difficult to be accurate about some comments)

Figures captions could specify the statistical test after p-value reporting. "* indicates significance (P < 0.05) by xxxx test." 

Figures 4 and 5 contain a lot of information, which can be confusing for the reader, and the letters are not legible (very small). 

Figure 9 caption needs to be edited. "Our objective was to predict ??? This is M&M

Many of the results are not discussed. Discussion requires to be improved including a more in-depth. For example, the conclusion indicates that rumen Ruminococcus can serve as distinguishable markers of calves' different HS responses. But the discussion about it is poor. 

Authors need to check the data presentation (tables and figures), choose the most important data for the main document, and put the other one into supplementary material. 

There are many typing errors that show the poor preparation of the manuscript, commas where there should be periods, spacing errors, etc. (Some pages do not have line numbers, which makes it difficult to be accurate about this comments)

Author Response

Comments 1: Please define the objective of the study in your introduction section. 

Response 1: We have made correction according to the Reviewer's comments (Lines 62-64).

Comments 2:  just 13 days is enough to select cows with different THI answers. Authors may please justify it?

Response 2: Our entire trial lasted a total of 51 days, but there was a lack of data due to frequent rain during the first 30 days, and minimum heat stress conditions were often not met during this period, so we kept collecting continuously. In addition, during the trial period shown in this paper, we also conducted a 7-day pre-experimental phase, but during this phase we only tested the calves' respiration rate, in order to allow the calves to acclimatize to heat stress for a period of time, rather than being in a period of acute heat stress to test for changes in the level of response of the calves. During the 13 days of the formal trial, we tested the calves' rectal temperature and respiratory rate every day, and we actually had 20 days of data on respiratory rate.

Comments 3: There is no Figure 1 and Figure 2. Figure 1 yet is cited on page 3.

Response 3: We are very sorry for our incorrect. We have made changes (all figures).

Comments 4: There are many typing errors that show the poor preparation of the manuscript, commas where there should be periods, spacing errors, etc. (Some pages do not have line numbers, which makes it difficult to be accurate about some comments)

Response 4: We are very sorry for our incorrect. We have checked all the contents of the full text and corrected the errors.

Comments 5: Figures captions could specify the statistical test after p-value reporting. "* indicates significance (P < 0.05) by xxxx test." 

Response 5: We have made correction according to the Reviewer's comments (Lines 295 and 335).

Comments 6: Figures 4 and 5 contain a lot of information, which can be confusing for the reader, and the letters are not legible (very small). 

Response 6: The data presented in the article is well thought out and in addition, we have resized and sharpened the images.

Comments 7: Figure 9 caption needs to be edited. "Our objective was to predict ??? This is M&M

Response 7: We have made correction according to the Reviewer's comments (Lines 439-440).

Comments 8: Many of the results are not discussed. Discussion requires to be improved including a more in-depth. For example, the conclusion indicates that rumen Ruminococcus can serve as distinguishable markers of calves' different HS responses. But the discussion about it is poor. 

Response 8: We have made correction according to the Reviewer's comments.

Comments 9: Authors need to check the data presentation (tables and figures), choose the most important data for the main document, and put the other one into supplementary material. 

Response 9: These data in the main document are data that we have thought about placing to better articulate some of the changes in physiology, blood, microbiology, and organismal metabolites of lactating calves in response to different heat stresses, and therefore we do not want to place these data in the supplemental document.

Comments 10: There are many typing errors that show the poor preparation of the manuscript, commas where there should be periods, spacing errors, etc. (Some pages do not have line numbers, which makes it difficult to be accurate about this comments)

Response 10: We are very sorry for our incorrect. We performed a full-text check and corrected any errors.

Reviewer 4 Report

Comments and Suggestions for Authors

This study explores variations in gut microbiota composition and serum metabolome among Holstein calves showing distinct responses to heat stress (HS) under identical environmental conditions. The research is well-structured and offers valuable insights into microbial and metabolic adaptations to HS in young calves. These findings deepen our understanding of how host-microbe interactions respond to thermal stress, potentially enhancing animal welfare and productivity. However, certain aspects require further clarity and enhancement:

  1. The discussion on alpha and beta diversity analyses is brief. A more thorough comparative analysis with other ruminant heat stress studies would provide deeper insights.
  2. The study identifies differentially abundant metabolites but lacks a robust connection to heat stress physiology. Further elaboration on the roles in stress adaptation is necessary.
  3. Titles for Figures 4 to 9 should be revised for clarity and conciseness.

Author Response

Comments 1: The discussion on alpha and beta diversity analyses is brief. A more thorough comparative analysis with other ruminant heat stress studies would provide deeper insights.

Response 1: We have made correction according to the Reviewer's comments (Lines 489-492).

Comments 2: The study identifies differentially abundant metabolites but lacks a robust connection to heat stress physiology. Further elaboration on the roles in stress adaptation is necessary.

Response 2: Regarding the differential metabolites identified in this study, it is currently evident based on the results available in the literature that these metabolites are related to energy metabolism that has been validated to respond, but there is no direct evidence as to what role they play specifically during heat stress in calves, and we are currently planning to study these differential metabolites in calves or in mice to see what effect they have on animals during heat stress.

Comments 3: Titles for Figures 4 to 9 should be revised for clarity and conciseness.

Response 3: Thank you for pointing this out. We've modified some of the things that didn't make sense to make it easier to understand.

Round 2

Reviewer 2 Report

Comments and Suggestions for Authors

The authors have addressed each of the points raised in the first revision. They only need to revise the sentence shown in L.23, as it appears incomplete.

Reviewer 4 Report

Comments and Suggestions for Authors

Accept in present form